# Systematic Literature Review of Food-Intake Monitoring in an Aging Population [note 1]

**DOI:** 10.3390/s19153265

**Published:** 2019-07-24

**Authors:** Enrique Moguel, Javier Berrocal, José García-Alonso

**Affiliations:** Av. de la Universidad, s/n. University of Extremadura, 10004 Cáceres, Spain

**Keywords:** monitoring food intake, aging population, SLR

## Abstract

The dietary habits of people directly impact their health conditions. Especially in elder populations (in 2017, 6.7% of the world’s population was over 65 years of age), these habits could lead to important-nutrient losses that could seriously affect their cognitive and functional state. Recently, a great research effort has been devoted to using different technologies and proposing different techniques for monitoring food-intake. Nevertheless, these techniques are usually generic but make use of the most innovative technologies and methodologies to obtain the best possible monitoring results. However, a large percentage of elderly people live in depopulated rural areas (in Spain, 28.1% of the elderly population lives in this type of area) with a fragile cultural and socioeconomic context. The use of these techniques in these environments is crucial to improving this group’s quality of life (and even reducing their healthcare expenses). At the same time, it is especially challenging since they have very specific and strict requirements regarding the use and application of technology. In this Systematic Literature Review (SLR), we analyze the most important proposed technologies and techniques in order to identify whether they can be applied in this context and if they can be used to improve the quality of life of this fragile collective. In this SLR, we have analyzed 326 papers. From those, 29 proposals have been completely analyzed, taking into account the characteristics and requirements of this population.

## 1. Introduction

At present, the world is experiencing population-aging, a trend that is both pronounced and historically unprecedented. Over the past six decades, countries had experienced only a slight increase in the share of people aged 65 years and older, from 8% to 10%. However, in the next four decades, this group is expected to rise to 22% of the total population, a jump from 800 million to 2 billion people [1].

This trend is even more worrying in rural regions, for example, Extremadura in Spain or Alentejo in Portugal. This kind of region has lower population density than the average, and they keep losing their young population to more socioeconomically developed regions. Therefore, they have a higher-than-average aged population while being economically disadvantaged with a fragile cultural and socioeconomic context. Additionally, due to low population density and youth migration to richer regions, elders frequently live alone [2].

Aging in these regions is not a problem by itself, at least not directly. However, as people become older, they are more prone to diseases such as cognitive impairment, diabetes, hypertension and cardiovascular problems. Different studies indicate that a significant number of these diseases related to aging have their origin in deficient nutrition [3].

Elders are a particularly disadvantaged group with respect to nutrition, especially if compared with the rest of the population in developed countries [3]. Frequently, the elderly suffer changes in their nutritional patterns that, in some cases, can cause significant damages to their physical condition. For instance, some older people change their dietary habits, increasing their intake of greasy and salty food, or decreasing the total ingested food. This change of patterns usually leads to important-nutrient losses that directly influence the health of older people [4]. Deficient eating habits can cause serious problems in the function and cognitive status of elders, in addition to a higher rate of mortality, such as due to cardiovascular problems or anorexia episodes [5].

During the last few years, important research efforts have been devoted to various aspects related to feeding, not only for aging populations but also for the general population, addressing different nutritional disorders. One of the main objectives of the scientific community is to precisely monitor and identify nutritional patterns and ingested by elderly people. Different studies have shown that elders’ nutritional patterns are a valid parameter for predicting their quality of life [6,7,8]. Specifically, there are approaches in this area to propose new algorithms, techniques or systems for improving food-intake monitoring.

Food-intake monitoring is intended to acquire information, such as the number of vitamins, minerals and other substances ingested by a person [9]. This information is then used for the identification of nutritional patterns and the detection of nutritional problems. Most works in this field focus on the state of nutrition and their relationship with different diseases, such as obesity [10], Alzheimer’s disease [11], depression [12], and metabolic syndrome [13].

In general, these approaches propose to periodically carry out different surveys to the elderly in order to know their food-intake patterns, what their illnesses are, and the evolution of both. In this process, the prevalent method of diet monitoring is manually recording survey results [14,15,16]. However, this is a tedious process that ends with a low adherence rate, reducing its long-term effectiveness [17,18]. To address the problem presented by manual recording, numerous technological solutions have been proposed. These solutions introduce the use of a wide range of devices, technologies, and algorithms to automatically identify different aspects of the food-intake process, like the type of food being eaten and the amount of ingested calories, or identifying the person ingesting the food.

However, food-intake monitoring solutions have additional difficulties when taking into account the circumstances of an aging population, particularly when the monitored elders live in rural environments like the ones mentioned above. The lack of infrastructure, the low average technical skills of people living on these regions, the loneliness of elders, and so forth, hinder the deployment and use of food-intake monitoring systems. A study of the existing proposals is necessary to know whether they can be deployed in these environments.

In this paper, a Systematic Literature Review (SLR) of current food-intake monitoring techniques is presented, focused on the coverage of the requirements of the elderly living in rural and low populated regions. This research is part of a European project (International Institute for Research and Innovation in Aging—0045-4IE-4-P) granted to improve the quality of life of elder people living in rural environments. This paper has been partly published in a 6-pages conference paper [19]. The main distinction between the two papers is that the conference paper reviewed some papers of interest on monitoring food-intake, while the current paper proposes an extensive and formal Systematic Literature Review. Section 1 and Section 2 have been modified and adapted to explain the problem and motivation for a systematic review of monitoring food-intake technology in an aging population. In this paper we include a new and extensive section on Section 3 and the explanation of each step covered in the review. A new section on the Section 4 of papers is then included in which a taxonomy is extracted. In Section 5, the data and projects evaluated have been updated. Other and different results were found than the paper presented at the conference. Finally, Section 6 and Section 7 are different because the results obtained in both papers are not the same.

To describe the survey that was carried out, the rest of the paper is structured as follows. Section 2 presents the motivation for monitoring the food-intake of aging populations in rural areas. Section 3 describes the whole SLR process in the five steps performed in this work to evaluate food-intake monitoring. Section 4 presents the obtained analysis of the proposals in the execution of the SLR. Section 5 details the different analyzed technological proposals. In Section 6 a discussion involving the considered approaches is provided. Finally, Section 7 concludes this work.

## 2. Motivation

Monitoring food-intake in an aging population is important for detecting nutritional problems and, thus, being able to prevent related diseases. As mentioned above, one of the cheapest and less intrusive ways to perform this monitoring is through technological solutions.

Existing monitoring systems usually focus on technical aspects related to automating the monitoring process and improving precision in food and intake detection. Aspects like overall user impression, social acceptance, or system outputs are often considered. However, most works do no take into account the specific context in which the systems are deployed. This is particularly relevant in the case of elders living in rural regions, since several aspects have to be taken into account to make them viable for these environments:**Poverty**. Rural regions are usually some of the most socioeconomically disadvantaged regions in developed countries. Therefore, the cost of a monitoring system is a relevant factor [20].**Technical skills**. In general, elder populations are not the most technically skilled demographic. Moreover, rural regions have lower-than-average literacy indices. Therefore, on average, elders living in these regions severely lack technical skills to use and manage these monitoring systems [21].**Infrastructures**. Rural regions also suffer from lack of infrastructure. These deficiencies range from technological infrastructures, like the lack of broadband or even stable internet connection, to basic infrastructures, like road or train access, or even access to fresh groceries [22,23].**Health professionals**. Related to the previous characteristic, these regions also have significant deficiencies in termsof health professionals and infrastructures. Small villages in these regions are usually far from hospitals or nursing homes, hindering inhabitant access to health professionals [24].**Loneliness**. All of the above contribute to elder loneliness. People living in these regions usually spend most of their time alone. This should be taken into account by any food-intake monitoring system since they cannot be supervised [25,26].

Works such as References [27,28] have already analyzed different food-monitoring technologies. These technologies were rated and described as taking mainly technical aspects into account. In this work, technological solutions were analyzed from a different perspective, specifically the adaptability of the proposed solutions to be used by elders living in rural areas.

The previously defined characteristics make some of the developed current technological solutions unusable in these environments. In this work, we evaluate the existing solutions from this point of view. In particular, the issues to review were as follows:**Intake detection**. Food-intake events are relevant for most monitoring systems. These events could be used to extract eating patterns or detect anomalies. Therefore, capabilities for intake detection of the proposed system are reviewed.**Type of food detection**. Not only are intake events relevant for a food-intake monitoring system but the ability to detect the type of food being ingested by the user and its quantity is also of vital importance to evaluate the nutrient intake of elderly people.**User identification**. In environments where the monitoring system may be used by more than one person, it is important that the system is able to identify the specific person who is eating in each moment.**Cost**. For rural regions, the economic cost of deploying or using a monitoring system is a factor could seriously impact on its acceptance. Therefore, we analyzed the cost of existing systems.**Supervision**. Some monitoring systems need the physical presence of a supervisor when used, either a health professional or a technician. As mentioned above, this requirement may present a problem when the system is used in rural regions. Therefore, the need for a supervisor to be present is evaluated.**Infrastructure needs**. Some monitoring systems depend on the presence of a certain infrastructure. It may be technical, like solutions that rely on a broadband internet connection, or it may be physical, like solutions that rely on intermediation of grocery stores. Due to the characteristics of rural regions, infrastructure needs of monitoring solutions are a relevant factor for their adoption.**Portability**. Related with the previous point, rural regions are usually not easy to reach. Therefore, a monitoring system should be portable enough to be moved to these regions and deployed in elderly living locations.

As far as the authors know, there are no works reviewing the suitability of food-intake monitoring systems to be deployed in rural regions. To simplify the problem, the researchers decided to consider an average profile of an elderly person with an intermediate level of literacy and an intermediate level of adaptation to technologies. Next, we describe the followed SLR process and evaluate the existing technological proposals for food-intake monitoring based on the above-mentioned features.

## 3. Methodology

As indicated above, the limitations of analysis were due to a rural environment with elderly people and with low population density, limitations such as the nonexistence of health centers and intermediaries or the difficulty in the use of infrastructures.

In addition, there are other limitations in this research such as database-searching restrictions, possible limitations in the keywords selected in this study or limitations in excluding articles in different languages.

With the aim of evaluating existing technological proposals for food-intake monitoring, we conducted an SLR [29,30,31] where we identified the main technologies that could be used for elderly living in rural environments.

In addition to achieving the above objectives, the SLR wants to classify the research papers in the field. In this way, we also sought to identify whether the scientific community has raised similar concerns to those identified in this work.

An SLR is a type of scientific research that aims to objectively and systematically integrate the results of empirical studies on a given research problem. Kitchenham et al. [30] and Petersen et al. [31] suggested a five-step procedure. The steps that were performed for this SLR are described below:a)**Define research questions**. The main goal of this SLR is to provide an overview of the research proposals on food-intake monitoring in an aging population, identifying the quantity and type of research and the results available within it. From specific research questions, we wanted to identity those proposals that best matched the requirements of elderly people living in a rural environment and the different issues in which higher effort should be devoted in increasing the user acceptance of these systems.b)**Carry out literature search**. Primary studies are identified by using search strings on scientific databases or by manually browsing through relevant conference proceedings or journal publications. A good way to create a search string is to structure it in terms of collection, intervention, comparison, and outcome [32]. The structures are driven by the research questions.c)**Select studies**. Inclusion and exclusion criteria are used to exclude studies that are not relevant to answering the research questions. For instance, although we wanted to analyze food-monitoring systems, those focused on animal monitoring could be excluded because their objectives and structure were completely different.d)**Classify articles**. We followed a systematic process composed of three phases: (1) only the title, abstract and keywords of each paper were analyzed in order to discard them or not, according to the inclusion and exclusion criteria; (2) the same elements of every paper were assessed in order to classify them according to issues defined in this work (Section 4); and (3), all papers were minutely analyzed in order to further refine the assessment.e)**Extract and analyze data**. Once the classification scheme was defined, the relevant papers were ordered by that scheme. In this step, we used an Excel sheet to document the data-extraction process. The table contains each category of the classification scheme. During analysis, after reviewing each paper, we provided a short rationale about why the paper should be in a certain category or in several categories. From the final table, the suitable technologies for being applied for monitoring elder people, the requirements that they met and the open issues that should be addressed to increase user acceptance were identified.

In the following, each of the five steps of this SLR are applied to the addressed research problem.


*a) Research Questions*


While the overall objective of this study could be summarized as understanding which techniques and technologies exist for food-intake monitoring, this objective is divided into three specific research questions to obtain more detailed knowledge and a comprehensive view of the topic.

The research questions to be answered in this study are the following:**RQ.** **1****Which techniques and/or technologies exist for food-intake monitoring?****RQ.** **2****Is there evidence on the technology interest for food control of the elderly in rural environments?****RQ.** **3****Which research areas and concrete problems in food-intake monitoring are the most addressed?**


*b) Data Sources and Search Strategy*


The search strategy was developed taking into consideration the specific terminology regarding food-intake monitoring. In order to perform the search, we took into account terms such as *food-intake*, *aliment monitoring* or *nutrition care*, and their main synonyms. We were also interested in the *detection*, *recording* or *sensing* (and their synonyms) of this food-intake. Finally, we were interested in the *techniques* and *technologies* used for this purpose. For these reasons, the search chain was systematically determined by carrying out a complete search on the research topic. The search string was as follows:


    "food monitoring" OR "food intake" OR "intake food" OR "food logging"



    OR "food nutrition" OR "food recognition" OR "foodstuff monitoring"



    OR "aliment monitoring" OR "dietary assessment" OR "dietary monitoring"



    OR "nutrition care"



    AND



    technology OR system OR implementation OR solution OR technique



    OR method OR application OR app OR tool OR framework OR architecture



    OR software OR algorithm



    AND



    sensor OR sensing OR detector OR detect OR detecting OR measurement



    OR measure OR monitor OR monitoring OR capture OR record OR store


The selected search engine was the Scopus database [33]. This library has a wide coverage of publications in the area of computer science (among others) and indexes several publications catalogues (including the MDPI, IEEE, ACM, Springer, and Elsevier libraries). Please note that only works published in conferences and journals in the area of computer science were taken into account (link to the search in Scopus [34]). Other databases, such as PubMed [35] and PsycInfo [36], were taken into account but because they did not offer detailed technical and technological results, the researchers decided to discard the search in those databases.


*c) Study Selection*


Two simple criteria were defined for the selection of studies:Inclusion. We considered all works about food-intake monitoring in people. We did not make a classification depending on technologies and techniques that were directly proposed for use in elderly populations. We evaluated every technology that could be used in this sector of the population.Exclusion. We decided to remove all those works that did not deal with food-intake in people. The same technologies or techniques were not used for monitoring people, or cattle and other animals. On the other hand, we also decided to discard all those works that were about the same technology and the same way of solving a problem, maintaining the original solution.

For logical reasons, duplicated articles were discarded.


*d) Paper Classification*


The issues identified in *Motivation* were used for organizing the papers. These problems define the main dimensions that should be considered to implement and deploy a valid solution in the environment described above.

To facilitate the understanding of the paper analysis and because of the high number of proposals, classification was carried out in three phases: (1) only the title, abstract and keywords of each paper were analyzed to discard them or not according to inclusion and exclusion criteria (detailed in the previous section); (2) the same elements of every paper were assessed in order to classify them according to the defined issues; and (3) full papers were analyzed in order to further refine the selected papers to thoroughly answer the defined research questions.


*e) Data Extraction and Synthesis*


For an SLR, Kitchenham et al. [30] and Petersen et al. [31] defined guides suggesting that other parts of the analyzed papers should only be evaluated in those cases in which they are not well-structured or are imprecise. For this research, we decided to completely analyze each full paper in order to better answer all the raised questions. Some important proposals could be left aside by only analyzing titles, abstracts and keywords.

Each defined step was carried out by the researchers signing this paper. In order to improve readability and to facilitate understanding the obtained results, we decided to focus only on explaining the paper classification (Step d) and analysis (Step e) in the sections below.

## 4. Paper Classification

In this section, the obtained papers from the search query are classified according to the issues defined above. As previously indicated, this step is divided into three phases. In each phase, an evaluation was obtained that helped understand the state of the art in food-intake monitoring.

### 4.1. Phase 1

From the execution of the previously defined search string [34] in Scopus, 326 papers were obtained, of which the title, abstract and keywords were evaluated. From those, 107 were rejected according to the established inclusion and exclusion criteria, that is, all works that were not related to food-intake (for example, References [37,38]), or regarding food-intake in animals were excluded (for example References [39,40]).

Fro papers in which there were doubts about their inclusion or exclusion, we decided to include them for a more exhaustive evaluation in the next phases of the review process.

Therefore, 219 papers were kept to be evaluated in the second phase.

### 4.2. Phase 2

Taking into account the classification/issues detailed in the Section 2 (intake detection, type of food detection, user identification, cost, supervision, infrastructure needs, and portability) and the relevance of these papers for our initial hypothesis, already presented in the research questions subsection; out of the 219 papers, 29 papers were identified as of interest to be evaluated in the third phase.

In order to improve the readability of the paper, information detailing the papers that were analyzed, the ones that were excluded and the exclusion reasons are not included in the paper. Nevertheless, that information can be found as an external resource in the following URL [41].

### 4.3. Phase 3

In this third phase, we proceeded to evaluate the remaining 29 papers. The full papers were analyzed in order to further refine the selected paper to thoroughly answer the defined research questions.

This evaluation allowed us to know the different existing technological proposals for food-intake monitoring. Instead of classifying the analyzed paper according to previously defined issues, we organized them depending on the technological approach in which their proposal was based. This allowed us to highlight the most relevant aspect of each monitoring technique. For some of them, this mean that the focus is on the used device since its features and limitations are the basis on which the monitoring is based on. For others, the focus is on software, for example, video based solutions that can rely on video recorded from different devices. In case of overlap we always chose the category to which more attention is paid in the corresponding paper. Similar classification approaches have been previously used in works like Reference [28] where wearable monitoring devices are classified by the approach they use for the monitoring or in Reference [42] where monitoring technologies are analyzed according to the device (PDAs) or the software used (image recording).

Further, the presented requirements are used to evaluate to what extent these approaches meet them. According to the scientific literature presented in this document, we could divide food-intake monitoring technologies into the following categories:**Smartphone**. This category includes approaches based on smartphone technologies, such as the device itself, mobile applications (APPs) or specific smartphone’s sensors. In this category are included those works whose main contribution is technology aimed at smartphones.**Image-based techniques**. Applications, techniques and/or algorithms that could obtain high-level understanding from digital images or videos. This category includes papers whose main contribution is technology that makes use of images or video to obtain data and information.**Wearables**. Solutions based on electronic devices that can be worn on the body, either as an accessory or as part of clothing material.**Smart Home**. This category incorporates advanced automation systems to provide elders with sophisticated monitoring and controlling systems over home functions.**IoT**. Approaches based on the *Internet of Things* (IoT) paradigm, such as Internet-connected devices with sensors. When an IoT device was designed to be used at home, we placed it in the Smart Home category.**Single Board Computers**. Technologies based on small computers, especially used for writing documents or processing small or light software systems. This type of technology is usually associated with low cost or low energy consumption projects.**Others**. Solutions that could not be included in the preceding categories (for example, web pages, algorithms, ontologies, etc.).

The different categories in the paper are clarified by providing some examples.

Table 1 shows the number of papers selected in Phases 2 and 3 and the number of paper classified in each of the technological categories presented above:

We can see that, in Phase 2, the result was 234 papers but there were 13 articles classified in two categories (for example, papers that deal with image-based technology using Smartphones); there was also one paper that could be classified into three categories (Image-based techniques, Smartphone and IoT, since it included sensors to send more information about food-intake to a mobile device). Finally, 219 papers were obtained (as explained in Phase 2 of this process). In the third phase, we considered each approach to only be in one category depending on the core technology used or applied. For example, if a work made use of smartphone and image-based technologies, but its core technology is a smartphone app, then we classified this proposal in the smartphone category.

On the other hand, we can see that, in phase 3, the result is 29 papers that were distributed along the different categories. Table 1 shows a summary of this distribution. Concretely, 45% of the papers were in the Wearable (24%) and Image-based (21%) categories.

Different approaches and technologies can be used to monitor the food-intake of elderly people living in rural environments but not all of them perfectly meet the defined requirements. In the following section, we thoroughly analyze the most important approaches and which issues they meet.

## 5. Data Extraction and Synthesis of Food-Intake Monitoring Techniques

### 5.1. Smartphone

From a Smartphone perspective, acceptance of mobile health (mHealth) technologies such as smartphone applications has experienced impressive growth, with over 259,000 mHealth apps available [43]. Fifty-eight percent of smartphones have downloaded a health-related app [44], fitness and nutrition being the most frequently downloaded type of app [45].

Smartphone apps, as presented in Reference [46], help professionals and private users in food monitoring but in all evaluated cases, information about ingested products is manually updated. This type of tool has two main advantages over existing traditional methods—it is easier to record every ingested food and automatic calculations on ingested calories can be performed. However, these methods do not make the data-collection process easier for the user, nor do they help increase the accuracy of food-intake estimation.

Other apps, such as those in References [47,48], make use of image-based techniques for identifying food-intake (making use of a client–server configuration). The user begins the process by taking images of food plates using their smartphone camera and images are then sent to the server for analysis. The analysis process consists of two main parts: food identification and volume estimation. Analysis results are then sent back to the smartphone for review and confirmation by the user. This greatly impairs elderly autonomy, as they require supervision and intermediaries.

On the other hand, there are other solutions based on image-based and smartphone technologies, such as [49], in which researchers propose a mobile real-time eating-action recognition system. The proposed system continuously recognizes users’ eating actions and estimates the categories of eaten food items during mealtimes. With this system, users or caregivers can know the total amount of eaten food, the amount of calories of eaten foods, even for the meals that they have not yet started to eat. The system can be deployed on a smartphone, continuously monitoring eating actions during mealtimes. It detects the moment when a user eats and extracts food regions near the user’s mouth and classifies them. Nevertheless, it only recognize five different kinds of ingredients, which greatly limits the proposed solution. In addition, the associated cost with this project is unknown.

Generally, this type of solution is usually unable to identify the person who carries out the food-intake and is in need of an intermediary.

### 5.2. Image-Based Techniques

Image-based techniques that do not use smartphones, for example, the online system known as FoodLog [50], rely on users taking pictures of their food dishes using a camera and sending them to the system by email. FoodLog uses image-analysis methods for food identification from which nutrients can then be calculated.

Similarly, PlateMate [51] allows users to upload food pictures and receive nutrition estimations within a few hours. These estimations consist of a list of foods in the photograph, with measures about serving size, calories, fat, carbohydrates and protein for each food item. Again, in this system, the user sends the images directly to the server and waits for the response. Therefore, it requires intermediaries, increasing its cost.

Along this line, Reference [52] proposed a first-person food-monitoring system to make food nutritional information available. In this work, researchers used human computation to identify eating moments in a first-person point of view by images taken by wearable cameras and recognizing eating moments by using image-based techniques. However, as in the previous case, this requires intermediaries. In addition, this work cannot detect intake moments.

On the other hand, there are studies such as Reference [53] that propose algorithms to automatically estimate food attributes (such as ingredients and nutritional values) by classifying the input image (sent via a mobile application). Researchers used different deep-learning models for accurate food identification. In addition to image analysis, attributes and ingredients are estimated by extracting semantically related words from a huge corpus of text collected over the Internet. In the same line, making use of deep-learning techniques and short-range depth camera, H. C. Liao et al. [54] proposed a system for estimating food-intake directly from the food tray. This simplifies food-intake measuring. Both cases, however, could not detect the moment of intake, nor could they identify the person who is eating.

We would also like to point out that there are approaches, such as Reference [55], complementing image-based techniques with context information. These researchers developed a context-based image-analysis system for dietary assessment to automatically segment, identify and quantify food items from images. In this work, the authors describe image-segmentation and object-classification methods used in their system to detect and identify food items and use the context information to refine the classification results. This contextual dietary information is provided by the user, either explicitly or implicitly, to correct potential classification errors. This means that the system must be supervised.

As happens in the smartphone section, this type of solution needs intermediaries, it is expensive and most of the time the user cannot be identified.

### 5.3. Wearable

From a wearable perspective, technology that is increasingly being used, several authors explored the use of chewing sounds captured through these devices to detect and characterize food-intake activity. Projects such as References [56,57,58,59,60,61,62] present implementation of acoustic ear-pad sensor devices to capture the vibrations of food chewing. Spectral sound analysis compares food-classification performance with food-texture clustering results for different foods. However, there is clearly a problem in the presence of loud vibrations, so it is challenging to discern food properties to identify food-intake. As all these systems have the same features and behave equally, for our study we used only one of them (reason for exclusion already defined in the SLR).

In the same line, there are specialized algorithms [63,64] to detect and characterize food-intake activity, making use of the chewing sounds. These algorithms achieved acceptable results for single-meal experiments in laboratory settings, where the number of consumed food types was restricted, but have not been evaluated in real settings. In addition, the associated cost is unknown.

Works like [65] (a wearable, mobile, and wireless necklace featuring an embedded piezoelectric sensor) or [66] (a wearable and wireless chest belt) identified food and drink-intake by detecting a person’s swallowing events. These systems are based on the key observation that a person’s otherwise continuous breathing process is interrupted by a short apnea when they swallow as part of the solid- or liquid-intake process using a wearable chest belt. These systems are only capable of detecting intake and do not differentiate between types of consumed food.

Other works, like References [67,68], used wearable sensors placed on the user’s hand in order to monitor food-intake. In Reference [67], the authors proposed an approach using an accelerometer sensor and video observations. Nevertheless, this study only identified if the user was eating rice or noodles. In Reference [68], the authors proposed a similar system but making use of a specific wrist-worn smartwatch.

In general, wearable projects can hardly detect the consumed food, and the cost of deploying these systems in real settings is unknown.

### 5.4. Smart Home

In the Smart Home category, there are projects such as Reference [69], performing an automatic acquisition of nutritional habits based on ambient-intelligence [70] techniques. The goal of these projects is to support nutritionists by providing an estimation of the nutritional habits of a user using smart-home technologies. For instance, Reference [69] is based on a smart tablecloth equipped with a finely grained pressure textile matrix and a weight-sensitive tablet. It allows spotting and recognizing food-intake-related actions, such as cutting, scooping and stirring, the identification of the plate/container on which the action is executed and the tracking of the weight change in the containers. In other words, it can determine how many pieces are cut on the main dish plate, how many are taken from the side dish, how many sips are taken from the drink, how fast the food is being consumed and how much weight is taken overall. In addition, the distinction between different eating actions, such as cutting, scooping, poking, provides clues to the type of consumed food and the way the meal is consumed. Again, this project requires supervision and intermediaries, and portability is not possible.

There are also household appliances such as Reference [71] (smart fridge) or Reference [72] (smart microwave oven) with applications for a smart household. These devices were designed for managing items stored inside them, advising their users with cooking methods depending on what kind of food is stored. They can also perform other functions such as dietary control, nutrition monitoring and eating-habit analysis. This type of project is interesting and helpful to monitoring foodstuff but not who consumes these products.

In general, these projects are very interesting but not useful for elderly people as they are not usually unsupervised, they do not identify the person who ingests the aliments, and the devices are not portable.

### 5.5. IoT

From an IoT perspective, projects are being developed for measuring food-nutrition facts through a pocket-sized nonintrusive near-infrared (NIR) scanner [73]. This device records NIR spectra reflected from foods and uses them as features to predict nutrients, for example, energy and carbohydrates. The researchers used partial least-squares (PLS) regression and support-vector regression (SVR) for prediction. This device can only infer percentages of food-nutrient contents. Therefore, to better monitor food-intake, it is crucial to provide food weight in order to calculate the exact nutritional content [74].

Making use of other technologies, there are cutlery IoT devices with different integrated sensors for food-intake monitoring, such as References [75,76]. In the specific case of Reference [75], the presented work is a spoon that can recognize meal composition during the intake process without user intervention or on-body instruments. This utensil makes use of the fact that light spectra reflected by foods are dependent on the food ingredients. By analyzing the reflected light spectra, the spoon can recognize which type of food is on top of the utensil. The proposal describes the prototype design and the food-recognition algorithm for 20 types of foods. On the other hand, Reference [76] presented another utensil, in this case, a knife device that can identify food. In this case, a small microphone is attached to the knife that records the cutting sound of the food. The system extracts spectrograms from cutting sounds and uses them as feature vectors to train a classifier, making use of the k-Nearest Neighbor method (k-NN), a support vector machine (SVM) and a convolutional neural network (CNN) to verify differences of the classification methods. Neither case specifies how to identify the specific person and the cost of these projects is unknown.

Other implementations, such as Reference [77], propose a hybrid model that relies on both data collected from sensors and participatory data collected from a social-network community. The model encompasses some key smart features including tracking food-intake, lifestyle and exercise activities, generating warnings and recommendations and triggering interventions whenever needed. The model also mines the collected data for statistical analysis that can be used by health authorities to have a clear picture of the health status of the population and might help in making rational and informed decisions. This type of project requires supervision and intermediaries.

In general, IoT projects cannot identify the person who ingests the food, and the associated cost is unknown.

### 5.6. Single Board Computers

Systems based on Single Board Computers have existed for a long time. They have the general operating features and nutrient databases needed for food-intake analysis. Eight of these single board computers were exhaustively analyzed in [78]. The different systems vary in cost, the number of foods, nutrients in the database and ease of food entry. The main difference with the projects analyzed above, however, is the low speed of analyzing food-intake and the complexity in learning and using single board computers for elderly people. They also require the supervision and management of an assistant for the elderly. Other examples of this type of single board computers are References [79] or [80].

As can be seen, all these projects are rather old or insufficient for the needs of this SLR and cannot be used by elder people.

### 5.7. Others

There are other projects that do not fit in the preceding sections by their typology, such as References [81,82,83].

In the case of Reference [81], the proposal describes a voice-based mobile nutrition-monitoring system integrating speech processing, natural-language processing (NLP), and text-mining techniques in a unified platform to facilitate nutrition monitoring. After converting spoken data to text, nutrition-specific data are identified within the text using an NLP-based approach. Furthermore, the information is combined with a tiered matching algorithm to search for the food name in a nutrition database and accurately compute different intake values (for example, type of food or calories). In this case, intermediaries are always required.

On the other hand, Reference [82] presents an expert system for a nutrition care process tailored for the specific needs of elders. Dietary knowledge is defined by nutritionists, encoded as nutrition-care-process ontology and used as an underlining base and standardized model for nutrition care planning. An inference engine was developed on top of the ontology, providing semantic reasoning infrastructure and mechanisms for evaluating the defined rules for assessing short- and long-term self-feeding behaviors of the elderly to identify unhealthy dietary patterns and detect malnutrition early. Although the system is more or less complete, it does not identify the moment of food ingestion. Furthermore, the proposal needs intermediate external processes and its portability is complex.

Finally, Reference [83] presents a secured web-based intuitive and interactive application to record personal information on mobile phones and other devices with an Internet connection. Information such as food-intake, blood pressure and any other records/health-monitor readings can be sent for remote monitoring by authenticated advisors. As a drawback, this type of project requires expert supervision.

## 6. Discussion

In the previous section, the analyzed proposals were categorized depending on the main technology used and the most important ones were described and evaluated. In this section, these works are compared and analyzed depending on the identified issues in Section 2. Figure 1 presents the number of proposals meeting each specific issue. This figure allows us to identify the issues that have received the most attention (green bar) and those in which fewer approaches have been proposed (red bar). Although the number of papers is not representative by itself of the relevance of a given issue, it does show the interest of the scientific community in these works, as previously mentioned. As can be seen, the most important issues or requirements met by the analyzed approaches are in the *infrastructure-need* (18 papers) and *intake-detection* (17 papers) categories. On the contrary, the issues less addressed by these approaches are the *food-type-detection* (17 papers) and *portability* (15 papers) categories.

Figure 1 is important in identifying which issues are more relevant to the scientific community, but it does not provide information about the requirements met by each specific approach. Table 2 shows this information detailing if each proposal meets each specific issue (**✓**) or not (**X**). Please note that, in some cases, a feature may only be met partially, or may be not specified in the article itself. This situation is indicated by symbol**—**Analyzing the results of this table, important insights were obtained:Related to smartphone technology, almost every approach detects food-intake and the type of food that the user is eating and is portable. Nevertheless, they do not detect the user that is eating, require supervision, and, most importantly, they require infrastructure to work. These are key requirements for deploying these techniques in rural environments.Regarding image-based proposals, most of them are able to identify the type of food that the user is eating, they are cheap to deploy and most of them are portable and do not require supervision. Nevertheless, they cannot detect the person who is eating and, again, they require an Internet connection to work.Wearable devices are able to identify food-intake, do not require supervision or infrastructure and they are portable. They cannot, however, identify the type of food the user is eating and, normally, they are expensive.Related to smart-home technologies, all of them can identify the type of food. Nevertheless, there are a lot of issues that these approaches do not address, such as user identification, they require supervision, and they are not portable.Regarding IoT proposals, as can be seen in the table, they more or less meet the same characteristics that are addressed by the proposal in the wearable category. This makes sense since they are quite related. However, while wearable proposals were capable of identifying food-intake, these approaches are focused on detecting the type of food that the user is eating. In addition, most of theses approaches cannot detect the user who is eating.Finally, approaches falling in both the Single Board Computers and Others categories are the ones that meet fewer features. They mainly focus on the identification of the type of food the user is eating.

Matching these technologies and the features that should be met for deploying these proposals in rural environments for monitoring elder people, smartphones, wearables and the IoT are the ones that address most issues. Nevertheless, most of them cannot identify the user who is eating. Furthermore, in the majority of cases, some infrastructure is needed (either a supermarket or an external web service for computing the obtained data).

One point of view that can be interesting is the realization of a study carrying out a holistic combination of two or more technologies treated in this revision.

## 7. Conclusions

One of the objectives of this study was to provide answers to the research questions (outlined in the beginning of the SLR).

**RQ.** **1**
**Which techniques and/or technologies exist for food-intake monitoring?**
There are numerous techniques and technologies for food-intake monitoring, of which we emphasize the original classification that we have outlined in this paper: Smartphone, Image-based techniques, Wearable, Smart Home, IoT, Single Board Computers and Others. Taking into account this classification and the obtained data, researchers are making more efforts in works using Image-based and Smartphone technologies.**RQ.** **2**
**Is there evidence on the technology interest for food control of the elderly in rural environments?**
There is a clear interest for food control of the elderly but it is still in an initial state and needs more efforts from all involved actors.**RQ.** **3**
**Which investigations and aspects are the most used and proposed?**
As the more relevant results, we can say that almost 50% of the papers in this phase are about Image-based techniques (29%—68 papers in the second phase of the SLR) and Smartphone technologies (20%—47 papers in second phase). These are the main trends that we observed throughout this study. There is also considerable contribution in IoT (18%—43 papers in the second phase of the SLR) and Wearable (16%—43 papers in second phase) areas. However, taking into account the most relevant ones for our research, we found that the most significant papers were Wearable (24%—7 papers in the third phase of the SLR) and Image-based techniques (21%—6 papers in third phase) technologies.

From the various studied solutions, gaps were identified for deployment in rural environments. None of the solutions meets all the established requirements. About the studied options, the best matches with the interests of elderly people in rural areas are:Making use of wearable technologies, the work of Reference [65] or Reference [66] are interesting because they can detect food-intake and identify the specific person without supervision and without intermediaries. However, these works cannot detect food in particular, and their prototypes have an expensive cost.In the Smart Home category, Reference [71] is very interesting due to the use of an everyday appliance such as a fridge. This device monitors the purchases and the food taken out, but can detect neither the intake moment nor identify the person. This solution is also not portable, which complicates its deployment in rural regions.The best options are found in the Smartphone category. The works of Reference [47] or Reference [48] developed a mobile application that send photos taken with smartphones to external web services and receives the extracted data from such photos. These solutions detect intake and food without supervision but they cannot identify the person who is eating. On the other hand, a project [46] presented another app, but on this occasion, online processing was not used. Instead, data insertion was carried out by remote dieticians and other healthcare professionals (supervised manner and with intermediaries).

Looking at the analyzed studies in the different fields in this SLR, we can say that a complete solution does not exist for monitoring food-intake in an aging population in rural environments with low-density population. Therefore, additional efforts are needed in the area.

Because of what we learned in this SLR, work should start by focusing on mobile technologies, as it is one that most older people already know, it is easy to deploy in rural environments and would not incur high costs for its implementation. Nevertheless, in order to meet the requirements, some open issues still need to be resolved:Most proposals using this technology cannot identify the user. New proposals, using a technique such as image recognition or analyzing the behavioral patterns stored in the mobile phone [84] to identify the user, are required.Mobile technologies usually rely on a cloud environment to process and store data. New food-intake monitoring techniques reducing the cloud-infrastructure requirements are needed, instead processing and storing the data in the mobile device itself. This, in some cases, also leads to a reduction of operating costs [85].The need to supervise treatment should also be reduced in order to decrease the cost and infrastructure requirements, and improve deployment of the developed solutions. Artificial intelligence can be used in this sense to reduce this aspect of the food-intake monitoring approaches.Finally, the requirements within this group can also vary according to the people profile (literacy level, technological skills, etc.). It would also be interesting to develop self-adaptive solutions that are able to change their behavior depending on the situation and profile of each person. This would improve the likelihood of success of these solutions, but it would also increase their development cost.

As future work, we will work on a new SRL focused on beverage intake. Both SLRs would allow us to identify the main open issues that should be addressed to deploy food-and-beverage intake-monitoring systems in rural environments. Currently, we are also working on mobile technology for monitoring and controlling the diet of the elderly.

## Figures and Tables

**Figure 1 sensors-19-03265-f001:**
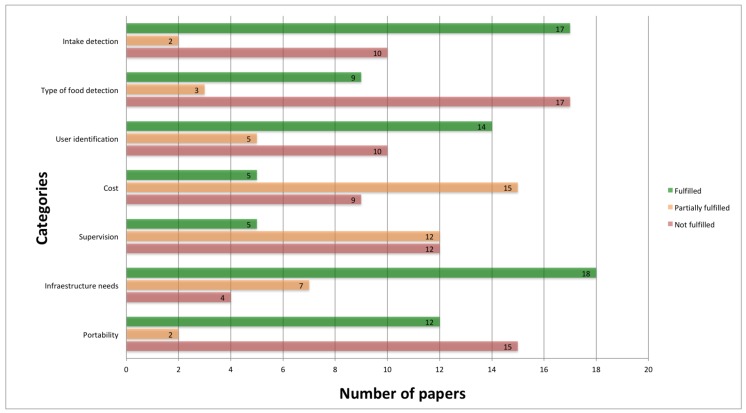
Number of papers in each category.

**Table 1 sensors-19-03265-t001:** Papers classification in phase 2 and phase 3.

Category	Phase 2	Phase 3
Smartphone	47	4
Image-based techniques	68	6
Wearable	36	7
Smart Home	7	3
IoT	43	4
Single Board Computers	2	2
Others	31	3
**Total**	**234**	**29**

**Table 2 sensors-19-03265-t002:** Comparative table of the selected papers for analysis.

Type	References	Intake Detection	Type of Food Detection	User Identification	Cost	Supervision	Infrastructure Needs	Portability
**Smartphone**	[46]	**—**	**✓**	**—**	**✓**	**—**	**—**	**✓**
	[47]	**✓**	**✓**	**X**	**—**	**—**	**X**	**✓**
	[48]	**✓**	**✓**	**—**	**—**	**X**	**X**	**✓**
	[49]	**✓**	**—**	**✓**	**—**	**✓**	**—**	**✓**
**Image-based techniques**	[50]	**X**	**✓**	**X**	**✓**	**X**	**X**	**X**
	[51]	**✓**	**✓**	**X**	**X**	**X**	**X**	**X**
	[52]	**✓**	**✓**	**—**	**—**	**✓**	**X**	**✓**
	[53]	**X**	**✓**	**—**	**✓**	**✓**	**X**	**✓**
	[54]	**X**	**✓**	**X**	**✓**	**✓**	**✓**	**—**
	[55]	**X**	**✓**	**X**	**—**	**✓**	**X**	**✓**
**Wearable**	[57]	**✓**	**—**	**—**	**—**	**✓**	**✓**	**✓**
	[63]	**X**	**—**	**—**	**X**	**X**	**X**	**X**
	[64]	**X**	**—**	**—**	**X**	**X**	**X**	**X**
	[65]	**✓**	**X**	**✓**	**—**	**✓**	**X**	**✓**
	[66]	**✓**	**X**	**✓**	**X**	**✓**	**✓**	**✓**
	[67]	**✓**	**X**	**—**	**—**	**✓**	**✓**	**✓**
	[68]	**✓**	**X**	**—**	**✓**	**✓**	**✓**	**✓**
**Smart Home**	[69]	**✓**	**✓**	**—**	**X**	**—**	**—**	**—**
	[71]	**X**	**✓**	**X**	**X**	**X**	**X**	**X**
	[72]	**X**	**✓**	**X**	**—**	**—**		**X**
**IoT**	[73]	**X**	**✓**	**X**	**X**	**✓**	**✓**	**✓**
	[75]	**✓**	**✓**	**X**	**—**	**✓**	**✓**	**✓**
	[76]	**X**	**✓**	**X**	**—**	**✓**	**✓**	**✓**
	[77]	**—**	**—**	**✓**	**X**	**X**	**X**	**✓**
**Single Board Computers**	[79]	**X**	**—**	**X**	**X**	**X**	**X**	**X**
	[80]	**X**	**✓**	**✓**	**—**	**X**	**X**	**X**
**Others**	[81]	**X**	**✓**	**X**	**—**	**✓**	**X**	**X**
	[82]	**X**	**—**	**—**	**—**	**—**	**X**	**X**
	[83]	**X**	**✓**	**—**	**—**	**X**	**X**	**✓**

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
