# Peer review of "Systematic Literature Review of Food-Intake Monitoring in an Aging Population"

_sensors, 2019, doi:10.3390/s19153265_

Reviewer 1 Report

In this paper the authors analyze the most important technologies and techniques proposed in order to identify if they can be applied in this context and if they can be used to improve the quality of life of this people. The topic is interesting and the contribution is well-defined. Although this work can be published I think the authors should improve some aspects:

- Add more quantitative information in the abstract.

- Define the limitations of this analysis.

- Explain more Figure 1 and Figure 2.

Author Response

We attach the complete response letter.

Reviewer 2 Report

Well structured paper about the need for food intake monitoring in elderly people, mainly in rural areas.

Paper is worth to be published even though some suggestions are provided to maximise impact and improve readability.

- Make paper to be reviewed by an English mother tongue, removing also some typos.

- Discuss about combined solutions using the different technologies. Maybe, an holistic approach  is worth to be proposed.

- Additionally, we are assuming that one solution can fit all the elderly in rural areas. Please, discuss about the chance to have different profiles of people (low or high digital literate, for example) and how can the solution in each case could be different.

- in the case of isolated people, please consider how monitoring food intake can alleviate isolation through different approaches (ie. volunteers, neighbours)

- Adherence is low for those monitoring activities if communication is happening in just one direction. Please, include another column in the study about how the different techniques or articles are providing feedback to the user. Adherence is just not a matter of technology but also about the added value for the citizen.

- Enlarge conclusions, specially from the perspective of future work. It's not enough to state that ' Therefore, additional efforts are needed in the area.'. Please, make explicit challenges to be solved

Author Response

We attach the complete response letter.

Reviewer 3 Report

The paper reports on a review of the food monitoring application to help older adults living in rural areas. The topic per se is interesting, however the authors do not report clearly why such narrow focus is so relevant. 

My main concerns are listed below:

- The paragraphs reporting the introduction and the motivation of the work are somewhat redundant. The work would be more attuned with an introduction that better focus the characteristics of the context of interest, better highlighting why it would be helpful for older adults living in rural areas to have easy access to food intake monitoring systems. The motivation paragraph should be revised underlining the features that are considered crucial for such systems deployed in the rural contexts.  

- The Authors limit their search to Scopus. However, given the topic, this is too narrow. They should extend it to include others relevant databases (e.g., PubMed, PsycInfo)

- Page 6 line 225 -> “We considered all those works about food intake monitoring in people (regardless of whether they were focused on elderly or not).” This inclusion criteria is wrong and inconsistent with what the Authors report at page 3, line 91. Technological applications are typically developed to target a specific population, which means they are designed to respond to the needs of a group of people. Older adults are a population group with specific characteristics, skills and needs, and considering all the works about food intake monitoring without focussing on the elderly, will not be of any help to really address the research question. The most likely outcome will be that most of the application are not suitable for older adults. 

- The paragraph reporting the methodology and the paper classification are redundant, reporting the process of your research referencing to the sources is enough.  

- Page 3 line91 -> “Technical skills. In general, elder population is not one of the most technically skilled demographics. Moreover, rural regions have lower than average literacy indexes. Therefore, on average, elders living on these regions would severely lack technical skills to use and manage these monitoring systems” The lack of technical skills does not mean that older adults should not be a target of intervention, rather it simply calls for an increased attention on the development of accessible tools.  

The classification of the different technologies is extremely weak. Authors should provide examples and also should motivate why categories are overlapping 

MINOR:

- Page 2 line 31 -> the sentence is not clear. Please revise

- Page 2 lines 32-38 -> these motivations highlights the importance of promoting behavior change applications to empower older adults to eat healthy 

- Page 2 line 44 - > of life[6], [7] and [8] : please revise 

- Page 4 line 145 -> [28] [29] & [30] : please revise

- Page 2 line 54 -> this is a tedious process that often end up : should be ends up

- Page line 56 -> …proposed. These solutions propose.. : please use synonyms 

- Figure 1 does not add any information, given Table 1

 Author Response

We attach the complete response letter.

Round  2

Reviewer 3 Report

I appreciated the Authors' effort to revise the manuscript as suggested. However, I still consider it unsuitable for publication as it is. 

I have three main points. The first regards the limitation of the literature review to only a few databases. This is not a limitation of the work, because it can be easily be accounted. Either the authors provide a sound motivation behind the choice, or they extend their search. As it is, they risk to omit relevant papers.  

Secondly, I still don’t find the categorization informative. Categories overlap in several cases and there is in fact no analysis on the true characteristics of the technologies deployed. For instance, ‘computer vision’ is a very broad label that, in turn, include many different technologies. The reason why categories overlap is that technological devices, i.e., a smartphone or wearable, are confounded with a technology, i.e, computer vision. Computer vision can be enabled by different technological devices, e.g., depth cameras embedded into the kitchen environment. 

The classification needs to be revised. 

In addition, now Authors claim that "This paper has been partly published in a 6-pages conference paper [19]." The authors should clearly underline how this extension of previous work makes a significant contribution to the community.

Finally, despite being revised the manuscript still contains many errors and typos. Proofreading is still needed.

Author Response

Thanks for your consideration and for your effort in reviewing and helping us improve the paper. In this new version, we have tried to address your concerns as best as possible in the available time. We believe that the paper is now better and we hope it meets your expectatives.

We attach the response letter, and when the editor allows us to upload the new paper, we will submit to MDPI platform.

Thank you very much.
